# Highly Sensitive and Specific Molecular Test for Mutations in the Diagnosis of Thyroid Nodules: A Prospective Study of *BRAF*-Prevalent Population

**DOI:** 10.3390/ijms21165629

**Published:** 2020-08-06

**Authors:** Yoon Young Cho, So Young Park, Jung Hee Shin, Young Lyun Oh, Jun-Ho Choe, Jung-Han Kim, Jee Soo Kim, Hyun Sook Yim, Yoo-Li Kim, Chang-Seok Ki, Tae Hyuk Kim, Jae Hoon Chung, Sun Wook Kim

**Affiliations:** 1Division of Endocrinology and Metabolism, Department of Medicine, Soonchunhyang University Bucheon Hospital, Bucheon 14584, Korea; yoonung2@hanmail.net; 2Division of Endocrinology and Metabolism, Department of Medicine, Korea University Ansan Hospital, Ansan 15355, Korea; psyou0623@gmail.com; 3Department of Radiology and Center for Imaging Science, Samsung Medical Center, Sungkyunkwan University School of Medicine, Gangnam-gu, Seoul 06351, Korea; helena35.shin@samsung.com; 4Department of Pathology, Samsung Medical Center, Sungkyunkwan University School of Medicine, Gangnam-gu, Seoul 06351, Korea; yl.oh@samsung.com; 5Division of Breast and Endocrine Surgery, Department of Surgery, Samsung Medical Center, Sungkyunkwan University School of Medicine, Gangnam-gu, Seoul 06351, Korea; junho.choe@samsung.com (J.-H.C.); jinnee.kim@samsung.com (J.-H.K.); js0507.kim@samsung.com (J.S.K.); 6BioSewoom Inc., Sungdong-gu, Seoul 04783, Korea; oksook@biosewoom.com (H.S.Y.); ylkim@biosewoom.com (Y.-L.K.); 7Green Cross Genome, Yongin 16924, Korea; changski.md@gmail.com; 8Division of Endocrinology and Metabolism, Department of Medicine, Thyroid Center, Samsung Medical Center, Sungkyunkwan University School of Medicine, Gangnam-gu, Seoul 06351, Korea; aledma623@gmail.com (T.H.K.); thyroid@skku.edu (J.H.C.)

**Keywords:** molecular diagnostic technique, fine-needle aspiration, thyroid cancer, *BRAF*-prevalent population, prospective study

## Abstract

Molecular testing offers more objective information in the diagnosis and personalized decision making for thyroid nodules. In Korea, as the *BRAF* V600E mutation is detected in 70–80% of thyroid cancer specimens, its testing in fine-needle aspiration (FNA) cytology specimens alone has been used for the differential diagnosis of thyroid nodules until now. Thus, we aimed to develop a mutation panel to detect not only *BRAF* V600E, but also other common genetic alterations in thyroid cancer and to evaluate the diagnostic accuracy of the mutation panel for thyroid nodules in Korea. For this prospective study, FNA specimens of 430 nodules were obtained from patients who underwent thyroid surgery for thyroid nodules. A molecular test was devised using real-time PCR to detect common genetic alterations in thyroid cancer, including *BRAF*, *N-*, *H-*, and *K-RAS* mutations and rearrangements of *RET/PTC* and *PAX8/PPARr*. Positive results for the mutation panel were confirmed by sequencing. Among the 430 FNA specimens, genetic alterations were detected in 293 cases (68%). *BRAF* V600E (240 of 347 cases, 69%) was the most prevalent mutation in thyroid cancer. The *RAS* mutation was most prevalently detected for indeterminate cytology. Among the 293 mutation-positive cases, 287 (98%) were diagnosed as cancer. The combination of molecular testing and cytology improved sensitivity from 72% (cytology alone) to 89% (combination), with a specificity of 93%. We verified the excellent diagnostic performance of the mutation panel applicable for clinical practice in Korea. A plan has been devised to validate its performance using independent FNA specimens.

## 1. Introduction

Thyroid nodules are the most prevalent endocrine disorders and are detected in approximately 50% of the adult population using high-resolution ultrasonography [1,2]. Fine-needle aspiration (FNA) cytology (FNAC) is the standard initial diagnostic approach and, using FNAC, most thyroid nodules are categorized as benign (Bethesda class II, 60–70% of cases), whereas approximately 5% are diagnosed as malignant (Bethesda class VI) [3,4]. Although the majority of thyroid nodules are classified as either benign or malignant, still approximately 20–30% of the cases are classified as indeterminate (Bethesda classes III, IV and V) using FNAC [3,4]. Of these indeterminate cytology cases, the risk of malignancy varies depending on whether noninvasive follicular thyroid neoplasms with papillary-like nuclear features (NIFTP) are considered as malignant (10–75%) or benign (6–60%) [3].

Recent advances in the understanding of the molecular pathogenesis of thyroid cancer have enabled the application of molecular tests to provide more objective information and to aid in clinical decision-making for personalized treatment. Since the first report of using *BRAF* V600E mutation analysis for FNA specimens by Kimura et al. [5], a “7-gene panel” consisting of *BRAF* V600E, *RAS*, *RET/PTC*, and *PAX8/PPARγ* mutations has been developed to overcome the low sensitivity of the testing for *BRAF* V600E single mutation and to increase the predictive value for malignancy, especially for indeterminate cytology from FNAC [6]. Mutation analysis of somatic mutations and gene fusions are considered as “rule-in” tests with their high specificity and positive predictive values. These tests have been used to predict thyroid cancer and to reduce the need for exploratory thyroid surgery. 

Mutation panels have been used mostly in the US population [7,8,9] and partly in Europe [10,11]. Nikiforov et al. and other groups reported improved sensitivity (48–86%) by using the 7-gene panel to detect the mutations in FNA indeterminate cytology specimens (three studies from the US [7,8,9], two from Italy [10,12], and one from Germany [11]). However, frequencies of these mutations in thyroid cancers differ among ethnicities, especially between Asian and Western populations. For example, *BRAF* V600E is the most prevalent mutation detected in papillary thyroid carcinoma (PTC), with an average frequency of 60–70% [13]; however, its frequency varies from 40 to 50% in the US [14] to over 80% in South Korea [15]. With the highest prevalence of *BRAF* V600E-associated thyroid cancer type in the world, a sole mutation test for *BRAF* V600E is commercially available for current clinical practice in Korea.

However, the clinical demand for precise cancer diagnosis in patients with indeterminate cytology has increased significantly in Korea. Therefore, we aimed to develop a 7-gene mutation panel for FNA specimens and to test its diagnostic performance in the Korean population, where *BRAF* V600E is the most prevalent mutation in thyroid cancer.

## 2. Results

### 2.1. Development of the Mutation Panel

A panel of mutations was assembled for the detection of the most common mutations in well-differentiated follicular cell-derived thyroid carcinomas. The panel was designed to detect *BRAF* V600E/K601E, *NRAS* codon 61 (Q61R, Q61L, Q61K), *HRAS* codons 12 (G12V), 13 (G13R), 61 (Q61K, Q61R), *KRAS* codons 12 (G12S, G12R, G12C, G12D, G12A, and G12V), 13 (G13D), 61 (Q61R) point mutations, *RET/PTC1*, *RET/PTC3*, and *PAX8/PPARγ* rearrangements. The mutation panel can directly analyze either DNA or RNA extracted from FNA specimens. The quality of nucleic acids isolated was considered satisfactory when the internal control (IC) cycle threshold (Ct) ≤ 34. With this criteria, 468 specimens (96.5%) demonstrated satisfactory quality of nucleic acid.

### 2.2. Detection of Mutations

Among the 430 FNA specimens, 293 (68.1%) were found to have genetic alterations using the mutation panel. The *BRAF* mutation (243 cases, 82.9%) was the most prevalent, followed by *RAS* mutation (33 cases, 11.3%), and *RET/PTC* rearrangement (17 cases, 5.8%) (Table 1). *PAX8/PPARγ* was not detected in this study population. Among the 243 *BRAF*-positive specimens, 241 cases had the *BRAF* V600E mutation, whereas two cases had *BRAF* K601E. Among the 33 *RAS*-positive specimens, 18 *NRAS* codon 61, six *HRAS* codon 61, four *KRAS* codon 12, and four *KRAS* codon 61 were identified using sequencing. One case had mutations for *NRAS* codon 61 and *HRAS* codon 61. Among the 243 *BRAF*-positive specimens, one case was found to have both *BRAF* V600E and *KRAS* codon 12 (G12C) mutations. This case was diagnosed as the classic type of PTC based on the reported solitary mass measuring 2.6 cm that extensively invaded the blood vessels and lateral lymph nodes. Among the 17 *RET/PTC* rearrangement-positive specimens, there were 16 *RET/PTC1* and one *RET/PTC3* mutations. All positive specimens for genetic alterations were confirmed by conventional sequencing, thus achieving a total agreement rate of 100%.

### 2.3. Analysis of All Specimens

All patients underwent thyroid surgery and 347 patients (80.7%) were diagnosed with thyroid cancer. Among the 430 FNA specimens, 251 (58.4%) were malignant (Bethesda class VI), 149 (34.7%) were indeterminate (Bethesda classes III, IV and V), and 30 (6.9%) were benign (Bethesda class II) (Figure 1). Among the 293 mutation-positive specimens, 287 cases (97.9%) were identified as thyroid cancer, including 15 NIFTP, whereas six mutation-positive specimens were confirmed as follicular adenoma (four with *RAS*-positive and two with *BRAF*-positive). A combination of cytology and molecular analysis improved the diagnostic performance, especially for sensitivity, which increased from 72.3% to 89.1%, while maintaining a high specificity of 92.8%, compared to that of cytology alone (Table 2).

Detection of each mutation was highly specific for cancer prediction. Detection of malignant rates for *BRAF*, *RET-PTC* and *RAS* were 99.2%, 100%, and 87.9%, respectively.

For malignant cytology (Bethesda class VI), molecular analysis was not necessary, because all cases were clinically diagnosed as cancer (Figure 1); thus, malignant cytology alone was sufficient to predict thyroid cancer in our institution. However, for benign cytology (Bethesda class II), a combination with molecular analysis reduced the false negative rate from 13.3% to 7.4%, although a 7% malignancy rate for benign cytology was not adequate to rule out diagnostic surgery.

### 2.4. Analysis of Indeterminate Specimens

Among the 149 FNA specimens with indeterminate cytology, 92 cases (61.7%) were diagnosed as thyroid cancer, including 15 NIFTP after surgery (Table 3). With molecular analysis, 61 cases (40.9%) were mutation-positive, among them, 56 cases (91.8%) were identified as cancer after surgery. Positive molecular results predicted thyroid cancer with a 60.9% sensitivity and a 91.2% specificity for indeterminate cytology. Considering that not all cancers harbor driver mutations for cancers, the performance of cancer molecular testing, especially in sensitivity, is rather limited. 

The prevalence of mutations in indeterminate cytology was distinctively different from that of the entire population with the *BRAF* as the predominant driver mutation for thyroid cancer. In our study on specimens of indeterminate cytology, *RAS* was the most prevalent mutation (29 cases, 47.5%), followed by *BRAF* (28 cases, 45.9%) (Table 3 and Figure 1). Most *RAS*-positive specimens were indeterminate cytology, except four cases (two with malignant cytology and two with benign cytology). Among the 33 *RAS*-positive results in the entire FNA specimens, *NRAS* (19 cases, 57.6%) was the most prevalent and 29 (87.9%) were diagnosed as cancer or NIFTP (Table 1); NIFTP was the most prevalent diagnosis (12 cases, 36.4%) in the *RAS*-positive specimens, followed by the invasive follicular variant of papillary thyroid carcinoma (FV-PTC, seven cases, 21.2%), classic follicular thyroid carcinoma (FTC, four cases, 12.1%), and classic PTC (three cases, 9.1%).

## 3. Discussion

We developed a 7-gene mutation panel for the qualitative evaluation of mutations, such as *BRAF* V600E, K601E, *NRAS* codon 61, *HRAS* codon 12, 13, 61, *KRAS* codon 12, 13, 61, *RET/PTC1*, *RET/PTC3*, and *PAX8/PPARγ*. The diagnostic performance of the 7-gene panel was comparable to the results obtained from Western populations. Diagnostic performance of the combined molecular testing and the cytology results was significantly enhanced in sensitivity 72% to 89% with a specificity of 93%, compared to cytology alone. All mutation-positive specimens, including *RAS*-positive, showed a 100% agreement with the results from Sanger sequencing, despite the diverse variations in *RAS* mutation. 

As expected, *BRAF* was the most prevalent mutation in the entire population of our study. *BRAF* V600E-positive nodules were predictive for classic PTC with >95% probability. Other types of PTCs, such as FV-PTC, Warthin-like variant of PTC, oncocytic variant of PTC, and diffuse sclerosing variant of PTC (DSV-PTC), also harbored the *BRAF* V600E mutation. Among the two *BRAF* K601E-positive cases, one was invasive FV-PTC and one was follicular adenoma, and *BRAF* K601E has been reported as a predictor for FV-PTC with a probability of 33–90% [16]. One *BRAF* V600E-positive case was detected by using real-time PCR and sequencing, and was confirmed as follicular adenoma after surgery. In the literature, false-positive results for *BRAF* V600E have been rarely reported, even for highly sensitive methods, including RT-PCR. Kim et al. reported six false-positive results in 126 specimens and, among them, *BRAF* V600E was also detected by using pyrosequencing in four cases [17]. Chen et al. reported five false-positive results in 292 specimens [18]. *RET/PTC* was also highly predictive for PTC (100%) and classic PTC (>82%). In our population, three DSV-PTCs were found to harbor *RET/PTC1*. Thus, *BRAF* V600E and *RET/PTC* confer > 99% probability of cancer, which is consistent with the literature [6,16]. Different from *BRAF* V600E and *RET/PTC*, *PAX8/PPARγ* is frequently detected in *RAS*-like tumors, such as FV-PTC and FTC [16]. In this study, we did not detect any *PAX8/PPARγ* mutation, although a considerable number of FV-PTC or FTC (51 cases, 11.8%) specimens were included. Song et al. reviewed the studies involving the Asian population, which reported a much lower prevalence of *PAX8/PPARγ* in thyroid cancers than those from the Western populations (on average, 6% in FTC from Asian vs. 44% from American *vs.* 27% from European) [19].

We found one case co-harboring *BRAF* V600E and *KRAS* G12C mutations, with extensive metastasis to the lateral lymph nodes (T3N1bM0). Generally, genetic alterations in *BRAF*, *RAS*, and *RET/PTC* are mutually exclusive in PTC. However, concomitant mutations in PTC have also been reported, especially in cases of advanced or recurrent PTC [20,21]. Zou et al. reported 11 cases (13%) harboring *BRAF* V600E and *KRAS*, or *BRAF* V600E and *RET/PTC1* mutations in 88 PTC specimens, and the majority were in the advanced stage [20].

*RAS* is the most prevalent mutation for indeterminate cytology, as highlighted in previous studies [22,23,24]. Harboring the *RAS* mutation presents different clinical meaning in that the accumulation of *RAS* is involved in the cancer clonal evolution from follicular adenoma to NIFTP or follicular carcinoma [25]. In contrast, a hyperplastic nodule is formed by a cluster of highly proliferative cells. Thus, the proportions of *RAS* or follicular adenomas included in the tumor populations could affect the diagnostic performance of the tested panel, especially for cases of indeterminate cytology. Compared to the earlier studies using the 7-gene panel (ThyroSeq v0) by Nikiforov et al. [8,9], our data show a higher rate of follicular adenoma in benign pathology and a higher rate of *RAS*-positive tumors for indeterminate cytology (Table 4). This could be an unfavorable condition for the diagnostic performance of the mutation panel and also for the fact that all tumor specimens included in this study were surgically resected. Nevertheless, our mutation panel produced a similar performance, compared to those from the previous studies using FNA specimens of indeterminate cytology; a sensitivity of 61% in this study, 61% from Nikiforov et al. involving 513 specimens [8], and 67% from Claudio Bellevicine et al. with 177 specimens [12]. In particular, the sensitivity (61–67%) and specificity (88–100%) of this mutation panel was comparable to the reported performance of the 7-gene panel in each Bethesda class, III, IV and V (sensitivity, 57–68% and specificity, 90–100%), except for the lower sensitivity (48%) of Bethesda class IV in this study. Although molecular testing improved the sensitivity, it still has limitations in sensitivity because not all tumors harbor genetic alterations that are cancer driver mutations [26]; in this study, approximately 16% of PTC cases were mutation-negative.

The probability of *RAS*-positive tumors for thyroid cancer was high in this study. Approximately 87% of tumors were diagnosed as thyroid cancer or NIFTP. Compared to the results from previous studies [24,27], NIFTP accounted for 41% of *RAS*-positive “carcinomas” in this study. Cancer prevalence of *RAS*-positive tumors varied between 12% and 100% [8,22,23,24,27], depending on the study cohorts and differences in the histology evaluation standard of each institution, especially for NIFTP (i.e. the presence of capsular invasion and nuclear features of PTC). In a recent multicenter study conducted by David et al. [27], 57% of tumors with *RAS*-like mutations were predicted for cancer or NIFTP. Interestingly, the prevalence of invasive FV-PTC was similar among the studies (21% in this study *vs.* 22% in David et al. [27]), although the prevalence of NIFPT ranged from 15% to 59% [24,27], indicating limitations in histological evaluation for NIFTP. The majority of *RAS*-positive tumors were of low risk, however, *RAS* was also detected in aggressive tumors; for example, approximately half of poorly differentiated thyroid carcinomas (PDTC) and a quarter of anaplastic thyroid carcinomas harbored the *RAS* mutation. In this study, 9% of the *RAS*-positive tumors were PTC variants with poorer prognosis with one each for the PDTC, DSV-PTC, and Warthin-like variant. In addition, NIFTP is regarded as an indolent tumor with very low risk of recurrence, but it can only be diagnosed after lobectomy due to its pathologic definition. Although *RAS*-positive tumors have a wide spectrum of tumor histology, from benign to malignancy, besides clinical suspicion and sonographic findings, molecular testing provides more objective information for diagnosis and clinical decision-making, such as lobectomy for nodules of indeterminate cytology.

In this study, we only included surgically resected tumors because pathology results are still the preferred gold standard and the diagnostic performance of the mutation panel used in this study has not been evaluated. Thus, the prevalence of cancer (81% in 430 cases) and the proportion of *BRAF* (83% in 293 mutation-positive specimens) were very high, compared to other studies (cancer prevalence of 15–35%) [8,10,11]. As our institution is a tertiary referral hospital in Korea, clinical decision in regard to thyroid surgery connotes the detection of tumors with the high probability of developing cancer, although the decision is made before molecular testing. For example, we reported in a previous study that among the 206 tumor specimens of Bethesda class III, 25% (50 tumors) were surgically resected, and 56% (28 tumors) were diagnosed as thyroid cancer [28]. Similarly, among the 68 tumor specimens with Bethesda class V, 63% (43 tumors) were surgically resected, and most of them (98%, 42 tumors) were confirmed as cancer [28]. Although cancer prevalence does not affect the sensitivity and specificity of the mutation panel, tumor profiles in this study could be somewhat different from those in the outpatient clinic in our institution or primary care providers. Thus, subsequent studies are required to validate the performance of this mutation panel with independent FNA specimens collected from outpatients during routine ultrasonography procedures.

## 4. Materials and Methods

### 4.1. FNA Specimens

We prospectively collected 485 FNA specimens from patients who underwent thyroid surgery at Samsung Medical Center between March 2014 and September 2016. This study was approved by the Institutional Review Board of Samsung Medical Center (IRB no. 2011-11-052) and was carried out in accordance with the Declaration of Helsinki. Informed consent was obtained from all patients included in the study. FNA specimens for the molecular testing were obtained on-site immediately after thyroid surgery by the surgeon, instead of the usual routine of ultrasonography-guided biopsy. Aspirated materials were placed into tubes containing 400 µL of RNAlater solution (Thermo Fisher Scientific, Inc., Wilmington, DE, USA) and the tubes were frozen and stored at −20 °C. All specimens were persevered for not longer than 6 months before molecular analysis. For cytology review, FNA specimens were obtained by the radiologist through ultrasonography procedures. To confirm whether the nodules of the removed tissue described in the surgical pathology report, or the FNA specimens for the molecular testing, corresponded to the nodules aspirated through the ultrasonography-guided procedures, all cases were reviewed independently by three endocrinologists (Y.Y.C., S.Y.P., and S.W.K.). Cases, in which the cytology report and features such as tumor size and location corresponded to the respective surgical pathology reports, were included in this study. We excluded 48 cases based on: specimens did not correspond to the reported nodules in the removed thyroid tissues, specimens with invalid molecular results, and specimens obtained from parathyroid tissue or diagnosed as medullar thyroid carcinoma (Figure 2). Thus, we included 437 FNA specimens from 437 independent thyroid nodules, and after excluding seven cases with nondiagnostic or unsatisfactory cytology review (Bethesda class I), a total of 430 cases were analyzed in this study.

### 4.2. Nucleic Acid Isolation

Total nucleic acid of specimens were isolated from the FNAC materials in the tubes using the QIAamp MinElute Virus Spin Kit (QIAGEN, Gaithersburg, MD, USA) according to the manufacturer’s instructions. The quantity and quality of isolated DNA and RNA was assessed using a NanoDrop 1000 spectrophotometer (Thermo Fisher Scientific, Inc., Wilmington, DE, USA). The isolated nucleic acids were frozen and stored at −70 °C. 

### 4.3. Detection of Mutations by Real-Time PCR

The mutation panel developed and used in this study can analyze either DNA or RNA mutations in a one-step assay without the step of complementary DNA synthesis. For real-time reverse transcription polymerase chain reaction (rRT-PCR), 25 ul aliquots contained 2 µl of the nucleic acids, x1 qRT–PCR mixture, 1 µl of qRT–PCR enzyme (Nanohelix, Daejeon, Korea) and optimized concentrations (0.1–0.4 µM) of primers and probes. A quantity of 4.8–393 ng of extracted nucleic acid was used for rRT-PCR. Nucleic acids were heated at 50 °C for 30 min for 1 cycle and 95 °C for 15 min for 1 cycle and 95 °C for 15 s, then subjected to 40 cycles of 60 °C for 45 s using an AB 7500 thermocycler (Thermo Fisher Scientific, Inc., Wilmington, DE, USA). The amplification of *KRAS*, *HRAS*, *RET/PTC1*, and IC was detected using a FAM-labelled assay. The amplification of *BRAF*, *NRAS*, *RET/PTC3*, and *PAX8/PPARγ* were detected using a VIC-labelled assay. The Ct of each gene was obtained at a threshold of 0.2 of FAM and VIC, using an AB 7500 thermocycler (Thermo Fisher Scientific, Inc., Wilmington, DE, USA). *RAS* mutation was regarded as positive when the ∆Ct value was ≤11. The ∆Ct value was calculated by [mutation assay Ct–IC assay Ct]. *RET/PTC* or *PAX8/PPARγ* mutation was regarded as positive when the Ct value was ≤39. Results analysis was only performed on specimens with IC Ct ≤ 34, which is a valid value of the assay. Specimens were excluded when the IC Ct was more than 34 cycles in the repeated assay.

### 4.4. Direct Nucleotide Sequencing

All mutation-positive specimens were sequenced using the Terminator v3.0 Cycle Sequencing Ready Reaction kit (Applied Biosystems, Foster City, CA, USA) on the ABI Prism 3130 Genetic Analyzer (Applied Biosystems, Foster City, CA, USA). We aimed for at least 95% compatibility of results between RT-PCR and sequencing.

### 4.5. Cytology Review

Routine cytological evaluation of all FNAC smear slides was conducted by one experienced cytopathologist (Y.R.O.) who was blinded to the results of molecular testing. Since the definition of NIFTP was introduced after the final description of the surgical pathology report for thyroid nodules included in this study, cases with FV-PTC were reviewed to classify them as either NIFTP or invasive FV-PTC.

### 4.6. Statistical Analysis

We calculated the sensitivity, specificity, positive predictive value, negative predictive value, and accuracy for the mutation panel using MedCalc statistical software v9.6 (MedCalc Software, Mariakerke, Belgium).

## 5. Conclusions

In this study, the high specificity and sensitivity of the mutation panel were verified for routine clinical oncology practices in *BRAF*-prevalent patient populations. We are collecting independent specimens to further validate the diagnostic performance of the mutation panel.

## Figures and Tables

**Figure 1 ijms-21-05629-f001:**
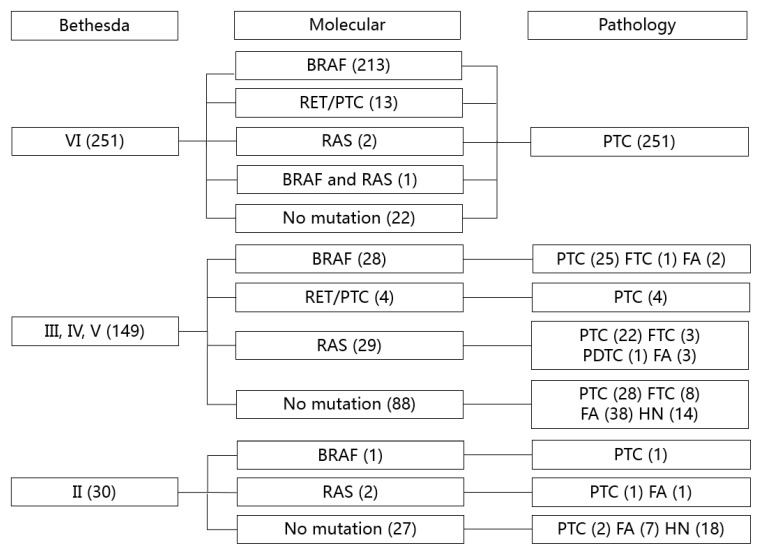
Correlation between cytology, molecular, and pathology diagnosis in 430 FNA specimens. FNA, fine-needle aspiration; PTC, papillary thyroid carcinoma; FTC, follicular thyroid carcinoma; PDTC, poorly differentiated thyroid carcinoma; FA, follicular adenoma; HN, hyperplastic nodule.

**Figure 2 ijms-21-05629-f002:**
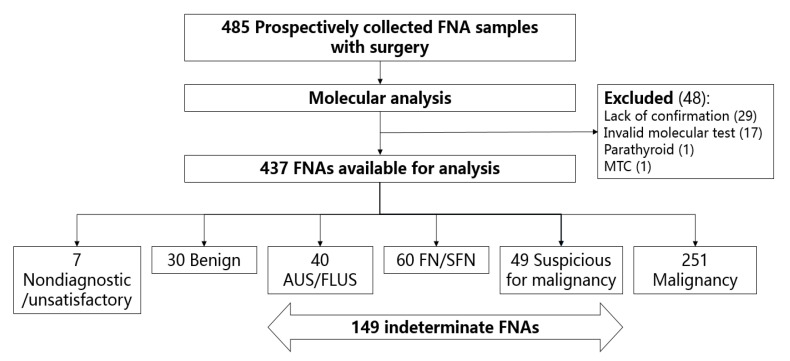
Schematic representation of the study design. FNA, fine-needle aspiration; MTC, medullary thyroid carcinoma; AUS/FLUS, atypia of undetermined significance/follicular lesion of undetermined significance; FN/SFN, follicular neoplasm/suspicious for follicular neoplasm.

**Table 1 ijms-21-05629-t001:** Molecular distribution and histological results.

Tumor Type	BRAF (242)		RET-PTC (17)		RAS (33)				Concomitant (1)
	BRAF V600E (240)	BRAF K601E (2)	RET-PTC1 (16)	RET-PTC3 (1)	NRAS (18)	KRAS (8)	HRAS (6)	HRAS and NRAS (1)	BRAF V600E and KRAS (1)
Classic PTC	228	-	13	1	3	-	-	-	1
Invasive FV-PTC	3	1	-	-	4	-	3	-	-
Classic FTC	1	-	-	-	2	1	1	-	-
DSV-PTC	1	-	3	-	-	1	-	-	-
Warthin-like variant of PTC	2	-	-	-	-	1	-	-	-
Oncocytic variant of PTC	1	-	-	-	-	-	-	-	-
PDTC	-	-	-	-	1	-	-		-
NIFTP	3	-	-	-	6	4	2	-	-
Follicular adenoma	1	1	-	-	2	1	-	1	-

PTC, papillary thyroid carcinoma; FV-PTC, follicular variant of papillary thyroid carcinoma; FTC, follicular thyroid carcinoma; DSV-PTC, diffuse sclerosing variant of papillary thyroid carcinoma; PDTC, poorly differentiated thyroid carcinoma; NIFTP, noninvasive follicular thyroid neoplasm with papillary-like nuclear features; FA, follicular adenoma.

**Table 2 ijms-21-05629-t002:** Comparison of diagnostic performance between cytology and molecular testing.

Diagnosis	Diagnostic Performance (%)				
	Sensitivity	Specificity	PPV	NPV	Accuracy
Cytology alone (Bethesda class VI)	72.3	100	100	46.4	77.7
Molecular test alone (mutation detected)	82.7	92.8	97.9	56.2	84.7
Cytology positive or molecular positive	89.1	92.8	98.1	66.9	89.8

PPV, positive predictive value; NPV, negative predictive value.

**Table 3 ijms-21-05629-t003:** Correlation between cytology, molecular, and pathology diagnosis in 149 indeterminate fine-needle aspiration (FNA) specimens.

**AUS/FLUS (40)**			
Mutation	Cancer or NIFTP (23)	Benign (17)	Molecular test Sensitivity 60.9% Specificity 88.2% PPV 87.5% NPV 62.5% Accuracy 72.5% Cancer prevalence 57.5%
Detected (16)	12 RAS [10 FV-PTC (7 NIFTP), 1 FTC, 1 PDTC] 1 BRAF (1 FTC) 1 RET-PTC (1 DSV-PTC)	2 BRAF (2 FA)	
Not detected (24)	9 [5 FV-PTC (2 NIFTP), 3 PTC, 1 FTC]	15 (9 FA, 1 oncocytic FA, 5 HN)	
**FN/SFN (60)**			
Mutation	Cancer or NIFTP (21)	Benign (39)	Molecular test Sensitivity 47.6% Specificity 92.3% PPV 76.9% NPV 76.6% Accuracy 76.7% Cancer prevalence 35%
Detected (13)	9 RAS [6 FV-PTC (3 NIFTP), 2 FTC, 1 Warthin-like variant of PTC] 1 BRAF (1 FV-PTC)	3 RAS (3 FA)	
Not detected (47)	11 [5 FV-PTC (1 NIFTP), 3 FTC, 3 oncocytic variant of FTC]	36 (22 FA, 6 oncocytic FA, 8 HN)	
**Suspicious for malignancy (49)**			
Mutation	Cancer or NIFTP (48)	Benign (1)	Molecular test Sensitivity 66.7% Specificity 100% PPV 100% NPV 5.9% Accuracy 67.4% Cancer prevalence 97.9%
Detected (32)	24 BRAF [21 PTC, 2 FV-PTC (1 NIFTP), 1 Warthin-like variant of PTC] 5 RAS [2 PTC, 2 FV-PTC (1 NIFTP), 1 DSV-PTC] 3 RET-PTC (3 PTC)	0	
Not detected (17)	16 (12 PTC, 3 DSV-PTC, 1 FTC)	1 (1 HN)	

FNA, fine-needle aspiration; AUS/FLUS, atypia of undetermined significance/follicular lesion of undetermined significance; FN/SFN, follicular neoplasm/suspicious for follicular neoplasm; NIFTP, noninvasive follicular thyroid neoplasm with papillary-like nuclear features; PTC, papillary thyroid carcinoma; FV-PTC, follicular variant of papillary thyroid carcinoma; FTC, follicular thyroid carcinoma; PDTC, poorly differentiated thyroid carcinoma; DSV-PTC, diffuse sclerosing variant of papillary thyroid carcinoma; FA, follicular adenoma; HN, hyperplastic nodule; PPV, positive predictive value; NPV, negative predictive value.

**Table 4 ijms-21-05629-t004:** Comparison of proportions of benign nodules and *RAS*-positive specimens between this study and previous studies.

Study	Bethesda Classes	Benign Histology		Mutation Detected	
		Hyperplastic Nodule	Follicular Adenoma	Total	RAS
Present study	III, IV, V	9.4% (14/149)	28.9% (43/149)	40.9% (61/149)	19.5% (29/149)
Nikiforov et al. (2009) (reference no. 9): 7-gene panel	III, IV, V	51.9% (27/52)	7.7% (4/52)	29.4% (15/51)	9.8% (5/51)
Nikiforov et al. (2011) (reference no. 8): 7-gene panel	III, IV, V	54.2% (278/513)	22.2% (114/513)	16.2% (83/513)	11.9% (61/513)

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
