# Peer review of "Highly Sensitive and Specific Molecular Test for Mutations in the Diagnosis of Thyroid Nodules: A Prospective Study of BRAF-Prevalent Population"

_ijms, 2020, doi:10.3390/ijms21165629_

Round 1

Reviewer 1 Report

The manuscript " Highly sensitive and specific molecular test for mutations in the diagnosis of thyroid nodules: A prospective study of BRAF-prevalent population." presents valuable data both from clinical and genetic perspective and certainly deserves publication. However, there are some important points that need to be addressed before it can be accepted for publication.

  1. In “Materials and Methods” section: Total nucleic acid of specimens were isolated from the FNAC materials in the tubes using the QIAamp MinElute Virus Spin Kit (Qiagen). I’m not sure what was the reason to choose extraction system that is dedicated to isolate viral DNA/RNA particularly from body fluids and not tissues. Authors should use adequate system for designed for tissues, that is containing certain enzymes mix (proteases, collagenases) and RNAlater compatible ensuring a high yield and quality of nucleic acids used for further analyses.
  2. “Detection of mutations” – authors designed a new real-time PCR test therefore all technical information and conditions should be included, allowing reproducibility of the study. The primer sequences for all mutations should be included as well as PCR enzyme used and PCR mixture content. The range of 4.8-393ng of template usage in PCR reaction is huge and should be avoided particularly in case of such high mutations-positive rate. Authors used nanodrop spectrophotometer so it was easy to use recommended amount of ~20 ng for every sample that is suitable for 25 ul PCR reaction, ensuring better specificity and sensitivity of the study. Regarding this, I think that in many cases confirmation of mutation presence using conventional sequencing method (far less sensitive) might be and should be more problematic. Authors have got 100% concordance in Sanger sequencing so in regard of analysis on cell mixture (without dissection) raise the question on criteria of analysis parameters of chromatograms (QC value of positive calls). In order to reliably confirm the efficiency of the test I would recommend to use commercially validated real-time PCR test (i.e. for BRAF) with internal controls of nucleic acid quantity and quality.
  3. Cytology review – The usage of real-time PCR sensitivity in tumor specimen depends on the cancer cell content in the tissue. Therefore, it is good that this examination has been done. Usually for majority of validated tests the limit is 5-10% of cells in the tissue. Therefore, authors should provide more information regarding this examination and aberrant cell content.

Reviewer 2 Report

This is an interesting and detailed prospective study on the usefulness of molecular tests in fine needle aspiration (FNA) cytological samples of thyroid nodules. The authors have used a panel of mutations that includes the most common molecular alterations in thyroid tumors, using real-time PCR. A few similar studies exist, but this study has been conducted in a population with a particularly high prevalence of BRAFV600E mutation in papillary thyroid carcinomas (PTC). 

Although most cases of PTC with BRAFV600E mutation are easily identified only with cytology, the authors have included in the study enough cases with negativity for BRAFV600E mutation. The authors show the utility of their molecular panel (specificity and sensitivity) in clinical practice on BRAF-prevalent patient populations. 

Author Response

Thank you for your encouraging feedback and careful review of this manuscript.